# On the DCI Framework for Evaluating Disentangled Representations: Extensions and Connections to Identifiability

**Cian Eastwood**[*,1,2]        **Andrei Liviu Nicolicioiu**[*1]        **Julius von Kügelgen**[*1,3]

**Armin Kekić**[1]        **Frederik Träuble**[1]        **Andrea Dittadi**[1,4]        **Bernhard Schölkopf**[1]

[1]Max Planck Institute for Intelligent Systems, Tübingen, Germany
[2]School of Informatics, University of Edinburgh
[3]University of Cambridge
[4]Technical University of Denmark

## Abstract

In representation learning, a common approach is to seek representations which disentangle the underlying factors of variation. Eastwood and Williams (2018) proposed three metrics for quantifying the quality of such *disentangled representations*: disentanglement (D), completeness (C) and informativeness (I). We provide several extensions of this DCI framework by considering *the functional capacity required to use a representation*. In particular, we establish links to identifiability, point out how D and C can be computed for black-box predictors, and introduce two new measures of representation quality: explicitness (E), derived from a representation's loss-capacity curve, and size (S) relative to the ground truth. We illustrate the relevance of our extensions on the `MPI3D-Real` dataset.

## 1 INTRODUCTION

A primary goal of representation learning it is to learn representations $r(x)$ of complex data $x$ that "make it easier to extract useful information when building classifiers or other predictors" (Bengio et al., 2013). *Disentangled* representations, which aim to recover and separate the underlying factors $z$ that generate the data as $x = g(z)$, are a promising step in this direction. In particular, it has been argued that such representations are not only interpretable (Kulkarni et al., 2015) but make it easier to extract useful information for downstream tasks by recombining previously-learnt factors in novel ways (Lake et al., 2017).

While there is no single, widely-accepted definition of disentanglement, many evaluation protocols have

---

[*]Equal contribution.

been proposed to capture different notions based on the relationship between the learnt representation or *code* $c = r(x)$ and the ground-truth data-generative factors $z$ (Locatello et al., 2020, Fig. 11). The metrics of Eastwood and Williams (2018)—*disentanglement* (D), *completeness* (C) and *informativeness* (I)—estimate this relationship by learning a *probe* $f$ to predict $z$ from $c$ (§ 2) and can be used to relate many other definitions and scores (Locatello et al., 2020, § 6).

In this work, we extend the DCI framework in several ways. Our main idea is that *the functional complexity or capacity required to recover $z$ from the learnt code $c$ is an important but under-explored aspect of evaluating representations.* For example, consider the extreme case of recovering some true label from either: (i) a noisy version thereof; or (ii) raw, high-dimensional data (e.g. images). Noisy labels will do quite well with just linear capacity, but are fundamentally limited by the noise corruption. In contrast, the raw data will likely do quite poorly with linear capacity, but will eventually outperform the noisy labels given sufficient capacity. As shown in Fig. 1, we find this behaviour to be exhibited by a broad range of representations and probes.

**Structure and contributions.** First, we connect the DCI metrics to two common notions of linear and nonlinear identifiability (§ 3), thereby establishing a link to the related field of independent component analysis. We then propose an extended DCI-ES framework (§ 4) in which we: (i) elucidate a means to compute D and C scores for arbitrary black-box probes $f$ (e.g., MLPs); and (ii) introduce two new complementary measures of representation quality that permit a more fine-grained evaluation—*explicitness* (E) or ease-of-use, derived from a representation's *loss-capacity curve* (see Fig. 2), and *size* (S) relative to $z$. In our experiments (§ 6), we illustrate the relevance of our extensions by comparing several different representations on the `MPI3D-Real` dataset (Gondal et al., 2019).

*Accepted for the Causal Representation Learning workshop at the 38[th] Conference on Uncertainty in Artificial Intelligence* (UAI CRL 2022).

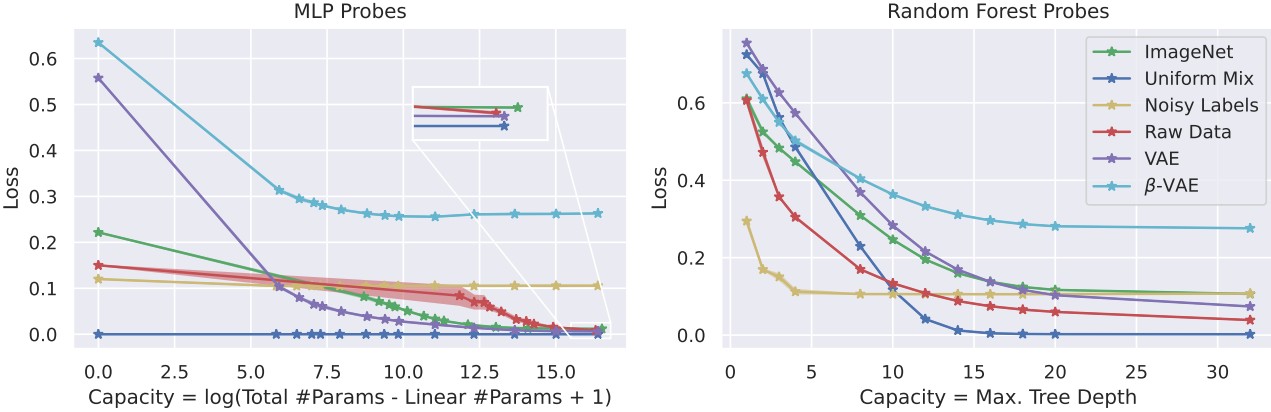

Figure 1: **Loss-capacity curves.** We show empirical loss-capacity curves (see § 4.2) for various representations (see legend) and two types of probes, multi-layer perceptrons (MLPs; left) and Random Forests (RFs; right), on the `MPI3D-Real` dataset. Results are averages over factors $z_j$, with means and 95% confidence intervals (shaded) computed over 3 random seeds.

## 2 BACKGROUND

Given a synthetic dataset of observations $x = g(z)$ along with the corresponding $K$-dimensional data-generating factors $z \in \mathbb{R}^K$, the DCI framework quantitatively evaluates an $L$-dimensional data representation or *code* $c = r(x) \in \mathbb{R}^L$ by: (i) training a probe $f$ to predict $z$ from $c$, i.e., $\hat{z} = f(c) = f(r(x)) = f(r(g(z)))$; and (ii) quantifying $f$'s prediction error and its deviation from the ideal one-to-one mapping, which is a permutation matrix (with extra "dead" units in $c$ whenever $L > K$).[1] For step (i), Eastwood and Williams (2018) use Lasso (Tibshirani, 1996) or Random Forests (Breiman, 2001) as linear or nonlinear predictors, respectively, for which it is straight-forward to read-off suitable "relative feature importances".

**Definition 2.1.** $R \in \mathbb{R}^{L \times K}$ is called a *matrix of relative importances* of $c$ for predicting $z$ via $\hat{z} = f(c)$ if $R_{ij}$ captures some notion of the contribution of $c_i$ to predicting $z_j$ such that for all $i, j$: $R_{ij} \geq 0$ and $\sum_{i=1}^{L} R_{ij} = 1$.

For step (ii), Eastwood and Williams use $R$ and the prediction error to define and quantify three criteria, or desiderata, of disentangled representations: *disentanglement* (D), *completeness* (C), and *informativeness* (I).

**Disentanglement.** Disentanglement (D) measures the average number of data-generating factors $z_j$ that are captured by any single code $c_i$. The score $D_i$ is given by $D_i = 1 - H_K(P_{i.})$, where $H_K(P_{i.}) = -\sum_{k=1}^{K} P_{ik} \log_K P_{ik}$ denotes the entropy of the distribution $P_{i.}$ over *row $i$* of $R$, with $P_{ij} = R_{ij}/\sum_{k=1}^{K} R_{ik}$. If

$c_i$ is only important for predicting a single $z_j$, we get a perfect score of $D_i = 1$. If $c_i$ is equally important for predicting all $z_j$ (for $j = 1, \ldots, K$), we get the worst score of $D_i = 0$. The disentanglement score $D$ is then given by the weighted average $D = \sum_{i=1}^{L} \rho_i D_i$, with $\rho_i = \frac{1}{K} \sum_{k=1}^{K} R_{ik}$.

**Completeness.** Completeness (C)[2] measures the average number of code variables $c_i$ required to capture any single $z_j$. The score $C_j$ in capturing $z_j$ is given by $C_j = (1 - H_L(\tilde{P}_{.j}))$, where $H_L(\tilde{P}_{.j}) = -\sum_{\ell=1}^{L} \tilde{P}_{\ell j} \log_L \tilde{P}_{\ell j}$ denotes the entropy of the distribution $\tilde{P}_{.j}$ over *column $j$* of $R$, with $\tilde{P}_{ij} = R_{ij}$. If a single $c_i$ contributes to $z_j$'s prediction, we get a perfect score of $C_j = 1$. If all $c_i$ equally contribute to $z_j$'s prediction, we get the worst score of $C_j = 0$. The overall completeness score is given by $C = \frac{1}{K} \sum_{j=1}^{K} C_j$.

**Informativeness.** The informativeness (I) of representation $c$ about data-generative factor $z_j$ is quantified by the prediction error, i.e., $I_j = 1 - \mathbb{E}[\ell(z_j, f_j(c))]$, where $\ell$ is an appropriate loss function. Here we deviate from Eastwood and Williams by defining $I_j$ such that 1 is the best score. The informativeness score is given by $I = \frac{1}{K} \sum_{j=1}^{K} I_j$.

**Remarks on the D and C scores.** *Together*, D and C quantify the degree of "mixing" between $z$ and $c$, i.e., the deviation from a one-to-one mapping. They are reported separately as they capture distinct criteria.

---

[1]W.l.o.g., it can be assumed that $z_i$ and $c_j$ are normalised to have zero mean and variance one for all $i, j$, for otherwise such normalisation can be "absorbed" into $g(\cdot)$ and $r(\cdot)$.

[2]also called *compactness* (Ridgeway and Mozer, 2018)

# 3 CONNECTION TO IDENTIFIABILITY

The goal of learning a data representation which recovers the underlying independent data-generating factors is closely related to blind source separation and independent component analysis (ICA) (Comon, 1994; Hyvärinen and Pajunen, 1999; Hyvarinen et al., 2019). Whether a given learning algorithm provably achieves this goal up to acceptable ambiguities, subject to certain assumptions on the data-generating process, is typically formalised using the notion of identifiability. Two common types of identifiability for linear and nonlinear settings, respectively, are the following.

**Definition 3.1.** We say that $c = r(x) = r(g(z))$ *identifies $z$ up to sign and permutation if $c = Pz$ for some matrix $P$ such that $|P|$ is a permutation matrix.*

**Definition 3.2.** We say that $c$ *identifies $z$ up to permutation and element-wise reparametrisation if there exists a permutation $\pi$ of $\{1, ..., K\}$ and invertible scalar-functions $\{h_k\}_{k=1}^{K}$ such that, for all $j$, $c_j = h_j(z_{\pi(j)})$.*

We now establish theoretical connections between the DCI framework and these types of identifiability.

**Proposition 3.3.** *If $D = C = 1$ and $K = L$ (i.e., $dim(c) = dim(z)$), then $R$ is a permutation matrix.*

*Proof.* First, by Defn. 2.1, we have $0 \leq R_{ij}$ and $\sum_{i=1}^{L} R_{ij} = 1$ $\forall i, j$, so $0 \leq R_{ij} \leq 1$. It follows that $\forall i, j : P_{i\cdot}, \tilde{P}_{\cdot j} \in \Delta_{K-1}$, where $\Delta_{K-1}$ denotes the $K$-dim. probability simplex, i.e., $P_{i\cdot}$ and $\tilde{P}_{\cdot j}$ are valid probability vectors. Hence, the Shannon entropies $H_K(P_{i\cdot}), H_K(\tilde{P}_{\cdot j})$ are well-defined $\forall i, j$, and, due to using $\log_K$ in the definition of $H_K$ (see § 2), are bounded in $[0, 1]$. It follows that $\forall i, j : 0 \leq D_i \leq 1$ and $0 \leq C_j \leq 1$. Since $D$ and $C$ are convex combinations of the $D_i$ and $C_j$, we have

$$D = 1 \iff \forall i : D_i = 1 \iff \forall i : H_K(P_{i\cdot}) = 0,$$
$$C = 1 \iff \forall j : C_j = 1 \iff \forall j : H_K(\tilde{P}_{\cdot j}) = 0.$$

Now for any $p = (p_1, ..., p_K) \in \Delta_{K-1}$, we have that

$$H_K(p) = -\sum_{k=1}^{K} p_k \log_K p_k = 0$$
$$\iff \forall k : p_k \log_K p_k = 0 \iff \forall k : p_k \in \{0, 1\}$$

where $p_k \log p_k := 0$ for $p_k = 0$, consistent with $\lim_{x \to 0^+} x \log x = 0$. Together with the simplex constraint, this implies that $p$ must be a standard basis vector $p = e_l$ for some $l$, i.e., $p_l = 1$ and $p_k = 0$ for $k \neq l$. Hence, $P_{i\cdot}, \tilde{P}_{\cdot j}$ must be standard basis vectors for all $i, j$, and so each row and column of $R$ contains exactly one non-zero element. Since columns of $R$ sum to one, these non-zero elements must all be one. $\square$

Using Prop. 3.3, we can establish links to identifiability, provided that the inferred representation $c$ perfectly predicts the true data-generating factors $z$, i.e., $I = 1$.

**Corollary 3.4.** *Under the same conditions as Prop. 3.3, if $z = W^\top c$ (so that $I = 1$) and $R = |W|$, then $c$ identifies $z$ up to permutation and sign (Defn. 3.1).*

*Proof.* By Prop. 3.3, $R$ is a permutation, so $W$ and thus $(W^\top)^{-1}$ must be signed permutation matrices. $\square$

For nonlinear $f$, we give a more general statement for suitably-chosen feature-importance matrices $R$.

**Corollary 3.5.** *Under the same conditions as Prop. 3.3, let $z = f(c)$ (so that $I = 1$) with $f$ an invertible nonlinear function, and let $R$ be a matrix of relative feature importances for $f$ (Defn. 2.1) with the property that $R_{ij} = 0$ if and only if $f_j$ does not depend on $c_i$, i.e., $||\partial_i f_j||_2 = 0$. Then $c$ identifies $z$ up to permutation and element-wise reparametrisation (Defn. 3.2).*

*Proof.* For any $j$ consider $z_j = f_j(c)$. By Prop. 3.3, $R$ is a permutation matrix, so column $j$ of $R$ contains exactly one non-zero entry in row $\pi(j)$ for some permutation $\pi$ of $\{1, ..., K\}$. Hence, by the assumed property of $R$, $f_j(c)$ does not depend on $c_i$ for all $i \neq \pi(j)$, and thus $z_j = f_j(c_{\pi(j)})$ $\forall j$. By invertibility of $f$, we obtain $c_j = h_j(z_{j'})$ with $h_j = f_{j'}^{-1}$ and $j' = \pi^{-1}(j)$. $\square$

*Remark* 3.6. While the *if* part of Corollary 3.5 holds for most feature importance measures, the *only if* part, in general, does not: not using a feature $c_i$ is typically a *sufficient* condition for $R_{ij} = 0$, but it need not be a *necessary* condition (as required for Corollary 3.5). E.g., measures based on *average* performance may not satisfy this since a feature may not contribute on average, but still be used—sometimes helping and sometimes hurting performance, see § 7 for further discussion.

Note that Gini importances, as used in random forests, *do* satisfy the necessary condition. While non-invertibility of random forests prevents an explicit link to identifiability (which is typically studied for continuous features), they can still be a principled choice in practice where factors are often categorical (see § 6).

# 4 EXTENDED DCI-ES FRAMEWORK

Motivated by our insights from § 3—considering different probe function classes provides different links to identifiability—as well the empirically-observed performance differences between representations trained with different-capacity probes (see Fig. 1), we now propose several extensions of the DCI framework.

## 4.1 PROBE-AGNOSTIC FEAT. IMPORTANCES

First, to meaningfully discuss more flexible probe function choices within the DCI framework, we point out that the D and C scores can be computed for arbitrary black-box probes *f* by using *predictor-agnostic* feature importance measures. In particular, in our experiments (§ 6), we use SAGE (Covert et al., 2020) which summarises each feature's importance based on its contribution to predictive performance, making use of Shapley values (Shapley, 1953) to account for complex feature interactions. Such predictor-agnostic measures allow D and C to be computed for probes with no inherent, built-in notion of feature importance (e.g., for MLPs), thereby generalising the Lasso and Random Forest examples of Eastwood and Williams (2018, § 4.3). While SAGE has several practical advantages over other probe-agnostic methods (see, e.g., Covert et al., 2020, Table 1), it may not satisfy the required conditions to link the D and C scores to different identifiability equivalence classes (see Remark 3.6). Future work may explore some of the alternative methods which do, e.g., by looking at a feature's mean *absolute* attribution value (Lundberg and Lee, 2017).

## 4.2 EXPLICITNESS (E)

We now introduce a new complementary notion of disentanglement based on the functional capacity required to recover or predict **z** from **c**. The key idea is to measure the *explicitness* or *ease-of-use* (E) of a representation using its *loss-capacity curve*.

**Setup.** Let $\mathcal{F}$ be a probe function class (e.g., random forests or MLPs), let $f_j^* \in \text{argmin}_{f \in \mathcal{F}} \mathbb{E}[\ell(z_j, f(\boldsymbol{c}))]$ be a minimum-loss probe for factor $z_j$ on a held-out data split[3], and let $\text{Cap}(\cdot)$ be a suitable capacity measure on $\mathcal{F}$—e.g., for random forests, $\text{Cap}(f)$ could correspond to the average tree-depth of $f$.

**Loss-capacity curves.** A loss-capacity curve for representation $\boldsymbol{c}$, factor $z_j$, and probe class $\mathcal{F}$ displays test-set loss against probe capacity for increasing-capacity probes $f \in \mathcal{F}$ (see Fig. 1). To plot such a curve, we must train $T$ predictors with capacities $\kappa_1, \ldots, \kappa_T$ to predict $z_j$, with

$$f_j^t \in \underset{f \in \mathcal{F}}{\text{argmin}} \, \mathbb{E}[\ell(z_j, f(\boldsymbol{c}))] \quad \text{s.t.} \quad \text{Cap}(f) = \kappa_t. \quad (4.1)$$

Here $\kappa_1, \ldots, \kappa_T$ is a list of $T$ increasing probe *capacities* ideally[4] shared by all representations, with suitable

---

[3]In practice, all expectations are taken w.r.t. the corresponding empirical (train/validation/test) distributions.

[4]True for RFs but not input-size dependent MLPs (§ 6).

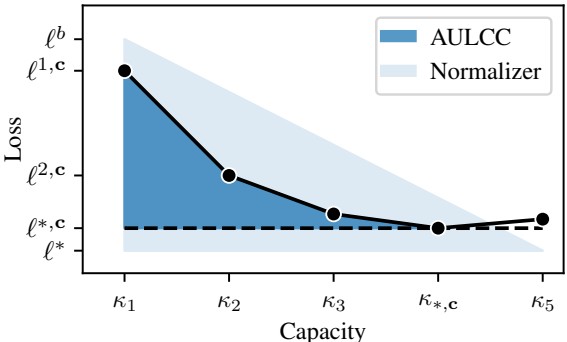

Figure 2: **Explicitness via the area under the loss-capacity curve (AULCC).** Here, $\kappa_1, \ldots, \kappa_T$ (x-axis) are a sequence of increasing function capacities and $\ell^{1,\boldsymbol{c}}, \ldots, \ell^{T,\boldsymbol{c}}$ (y-axis) are the losses achieved by the corresponding optimal predictors for $\boldsymbol{c}$. The lowest loss $\ell^{*,\boldsymbol{c}}$ is achieved at capacity $\kappa_{*,\boldsymbol{c}}$, while $\ell^b$ and $\ell^*$ are suitable baseline and best-possible losses for the probe class.

choices for $\kappa_1$ and $\kappa_T$ depending on both $\mathcal{F}$ and the dataset. For example, we may choose $\kappa_T$ to be large enough for all representations to achieve their lowest loss and, for random forest $f$s, we may choose $\kappa_1 = 1$ and then $T - 2$ tree depths between 1 and $\kappa_T$.

**AULCC.** We next define the *Area Under the Loss-Capacity Curve* (AULCC) for representation $\boldsymbol{c}$, factor $z_j$, and probe class $\mathcal{F}$ as the (approximate) area between the corresponding loss-capacity curve and the loss-line of our best predictor $\ell_j^{*,\boldsymbol{c}} = \mathbb{E}[\ell(z_j, f_j^*(\boldsymbol{c}))]$. To compute this area, depicted in Fig. 2, we use the trapezoidal rule

$$\text{AULCC}(z_j, \boldsymbol{c}; \mathcal{F}) = \sum_{t=2}^{t^{*,\boldsymbol{c}}} \left( \frac{1}{2} \left( \ell_j^{t-1,\boldsymbol{c}} + \ell_j^{t,\boldsymbol{c}} \right) - \ell_j^{*,\boldsymbol{c}} \right) \cdot \Delta\kappa_t,$$

where $t^{*,\boldsymbol{c}}$ denotes the index of $\boldsymbol{c}$'s lowest-loss capacity $\kappa_{*,\boldsymbol{c}}$; $\ell_j^{t,\boldsymbol{c}} = \mathbb{E}[\ell(z_j, f_j^t(\boldsymbol{c}))]$ the test-set loss with predictor $f_j^t$, see Eq. (4.1); and $\Delta\kappa_t = \kappa_t - \kappa_{t-1}$ the size of the capacity interval at step $t$. If the lowest loss is achieved with the lowest capacity ($t^{*,\boldsymbol{c}} = 1$), we set AULCC $= 0$.

We now define the **explicitness** (E) of representation $\boldsymbol{c}$ for predicting factor $z_j$ with predictor class $\mathcal{F}$ as

$$E(z_j, \boldsymbol{c}; \mathcal{F}) = 1 - \frac{\text{AULCC}(z_j, \boldsymbol{c}; \mathcal{F})}{\frac{1}{2}(\kappa_T - \kappa_1)(\ell_j^b - \ell_j^*)},$$

where $\ell_j^b$ is a suitable baseline loss (e.g., that of $\mathbb{E}[z_j]$) and $\ell_j^*$ a suitable lowest loss (e.g., 0) for $\mathcal{F}$. Here, the denominator represents the area of the light-blue triangle in Fig. 2, *normalizing* the AULCC such that $E \in [-1, 1]$ so long as $\ell_j^* < \ell_j^b$. The best score $E = 1$ means that the best loss was achieved with the lowest-capacity

probe $f_j^1$, i.e., $\ell_j^{*,\mathbf{c}} = \ell_j^{1,\mathbf{c}}$ and $\kappa_{*,\mathbf{c}} = \kappa_1$, and thus our representation $\mathbf{c}$ was explicit for predicting $z_j$ with $f \in \mathcal{F}$ since there was *no surplus capacity required (beyond $\kappa_1$) to achieve our lowest loss*. In contrast, $E = 0$ means that the loss reduced *linearly* from $\ell_j^b$ to $\ell_j^*$ with increased probe capacity, i.e., AULCC $=$ Normalizer in Fig. 2. More generally, if $\ell^{*,\mathbf{c}} = \ell^*$, i.e. the lowest loss for $\mathcal{F}$ can be reached with $\mathbf{c}$, then $E < 0$ implies that the loss decreased *sub-linearly* while $E > 0$ implies it decreased *super-linearly*. The overall explicitness score is simply the arithmetic mean over the $K$ factors, i.e., the $z_j$s.

**E vs. I.** While the informativeness score $I_j$ captures the (total) amount of information in $\mathbf{c}$ about $z_j$, the explicitness score $E_j$ captures the ease-of-use of this information. In particular, while $I_j$ is quantified by the *lowest prediction error with any capacity $\ell^{*,\mathbf{c}}$*, corresponding to a single point on $\mathbf{c}$'s loss-capacity curve, $E_j$ is quantified by the *area under this curve*.

**A fine-grained picture of identifiability.** Compared to the commonly-used mean correlation coefficient (MCC) or Amari distance (Amari et al., 1996; Yang and Amari, 1997), the $D, C, I, E$ scores represent empirical measures which: (i) easily extend to mismatches in dimensionalities, i.e., $L > K$; and (ii) provide a more fine-grained picture, for if the initial probe capacity $\kappa_1$ is linear and $R$ satisfies Corollary 3.5, we have that:

- $D = C = I = E = 1 \implies$ up to sign and permutation.
- $D = C = I = 1 \implies$ up to permutation and element-wise reparametrisation.
- $I = E = 1 \implies$ up to an invert. linear transformation.

Thus, if $D = C = I = E = 1$ does not hold exactly, which score deviates the most from one may provide valuable insight into the type of identifiability violation.

**Probe classes.** As emphasized throughout this section, whether or not a representation $\mathbf{c}$ is "explicit" for predicting factor $z_j$ depends on the class of probe $\mathcal{F}$ used, e.g., random forests or *multi-layer perceptrons* (MLPs). More generally, the explicitness of a representation depends on the way in which it is used in downstream applications, with different downstream uses or probes resulting in different definitions of explicit or easy-to-use information. We thus conduct experiments with different probes in § 6.

### 4.3 SIZE (S)

We next introduce a measure of relative *size* (S):

$$S = \frac{K}{L} = \frac{\dim(\mathbf{z})}{\dim(\mathbf{c})}.$$

When $L \geq K$, which is usually the case in representation learning, we have $S \in (0, 1]$ and a perfect score of $S = 1$. However, if we also consider the $L < K$ case, which would likely sacrifice some informativeness, we have $S \in (0, K]$. As we discuss in § 6, increased representation size often improves other scores like $I$ and $E$. Reporting $S$ thus permits an analysis of this size-informativeness or size-explicitness trade-off.

## 5 RELATED WORK

**Explicit representations.** Eastwood and Williams (2018, § 2) noted that the informativeness with a linear probe quantifies the amount of information about $\mathbf{z}$ in $\mathbf{c}$ that is "explicitly represented", while Ridgeway and Mozer (2018, § 3) proposed a measure of "explicitness" which simply reports the informativeness score with a linear probe. In contrast, our DCI-ES framework differentiates between the amount of information about $\mathbf{z}$ in $\mathbf{c}$ (*informativeness*) and the ease-of-use of this information (*explicitness*). This allows a more fine-grained analysis of the relationship between $\mathbf{z}$ and $\mathbf{c}$, both theoretically (distinguishing between more identifiability equivalence classes; § 3) and empirically (§ 6).

**Loss-capacity curves.** Plotting loss as a function of model complexity or capacity has long been used in statistical learning theory, e.g., for studying the bias-variance trade-off (Hastie et al., 2009, Fig. 7.1). More recently, such loss-capacity curves have been used to study the double-descent phenomenon of deep neural networks (Belkin et al., 2019; Nakkiran et al., 2021). However, they have yet to be used for assessing the quality or explicitness of representations.

**Loss-data curves.** Whitney et al. (2020) use loss-data curves, which plot loss against dataset size, to assess representations. They measure the quality of a representation by the *sample complexity* of learning probes that achieve low loss on a task of interest. In contrast, we focus on *functional complexity* and the task of predicting the data-generative factors $\mathbf{z}$, then discuss the functional complexity for other tasks $\mathbf{y}$ in § 7.

## 6 EXPERIMENTS

**Data.** We use `MPI3D-Real` (Gondal et al., 2019), a common disentanglement dataset containing $\approx$ 1M real-world images of a robotic arm holding different objects with seven annotated ground-truth factors: object colour (6), object shape (6), object size (2), camera height (3), background colour (3) and two degrees of rotations of the arm ($40 \times 40$); numbers in brackets indicate the number of possible values for each factor.

**Representations.** We use the following synthetic baselines and standard models as representations:

- *Noisy labels:* $c = z + \epsilon$ with $\epsilon \sim \mathcal{N}(\mathbf{0}, 0.01 \cdot \mathbf{I}_K)$.
- *Uniform mix:* $c = Wz$, where $W_{ij} = \frac{1}{LK} + \epsilon_{ij}$ with $\epsilon_{ij} \sim \mathcal{N}(0, 0.0016)$ (to ensure invertibility of $W$ a.s.).
- *Raw data:* $c = x$.
- *Others:* We also use VAEs (Kingma and Welling, 2014) with 10 latent variables ($L{=}10$), $\beta$-VAEs (Higgins et al. 2017, $\beta{=}100$, $L{=}10$); and an ImageNet-pretrained ResNet18 (He et al. 2016, $L{=}512$).

**Probes.** We use different MLP and RF probes $f$ to predict $z$ from $c$, with MLPs allowing us to analyse ease-of-use with neural networks. For MLPs, we start with linear probes (no hidden layers) then increase capacity by adding two hidden layers and varying their widths from $2 \times K$ to $512 \times K$. We then measure capacity based on the number of "extra" parameters beyond that of the linear probe, and compute feature importances using SAGE with permutation-sampling estimators and marginal sampling of masked values (https://github.com/iancovert/sage). For RFs, we use ensembles of 100 trees, control capacity by varying the maximum allowed depth between 1 and 32, and compute feature importances using Gini importance.

**Implementation details.** We split the data into training, validation and testing subsets of size 295k, 16k, and 726k respectively. We use the validation split for hyperparameter selection and report results on the test split. We train the MLP probes using the Adam (Kingma and Ba, 2015) optimizer for 100 epochs. We use mean-square error and cross-entropy losses for continuous and discrete factors $z_j$, respectively. To compute $E$, we use $\mathbb{E}[z_j]$ or a random classifier as baseline losses for continuous and discrete $z_j$, respectively.

**Results.** Fig. 1 depicts loss-capacity curves, averaged over factors $z_j$. For both probe types, the noisy labels baseline performs quite well with low-capacity probes but is surpassed by other representations given sufficient capacity, as expected. The uniform mix baseline is explicit for MLP probes (achieving $\approx 0$ loss with linear capacity) but not for RF probes, supporting the idea that the explicitness or easy-of-use of a representation depends on the way in which it is used. Note that, with MLP probes and $\log(\text{excess \#params})$ as the capacity measure, larger input representations are afforded more parameters with a linear probe and thus are more expressive. This highlights the difficulty of measuring the capacity of MLPs—an active area of research in its own right, which we discuss in § 7.

Table 1 reports the corresponding full DCI-ES scores, along with some oracle scores for MLPs. We find that:

Table 1: **DCI-ES scores for different probes and representations.** We report empirical scores using MLP and random forest (RF) probes trained on the `MPIP-3D` dataset, as well as theoretical/oracle for MLPs (MLP*) for some simple representations. We show averages over 3 random seeds; standard deviations were all $< 0.02$.

| Representation | Probe | D | C | I | E | S |
|---|---|---|---|---|---|---|
| GT Labels $z$ | MLP* | 1 | 1 | 1 | 1 | 1 |
| Noisy labels | MLP* | 1 | 1 | 0.9 | 1 | 1.0 |
| | MLP | 0.97 | 0.97 | 0.89 | 0.99 | 1.0 |
| | RF | 0.75 | 0.76 | 0.89 | 0.98 | 1.0 |
| Uniform mix | MLP* | 0 | 0 | 1 | 1 | 1.0 |
| | MLP | 0.13 | 0.22 | 1.0 | 1.0 | 1.0 |
| | RF | 0.17 | 0.21 | 1.0 | 0.72 | 1.0 |
| VAE | MLP | 0.15 | 0.14 | 0.99 | 0.71 | 0.7 |
| | RF | 0.10 | 0.10 | 0.93 | 0.65 | 0.7 |
| $\beta$-VAE | MLP | 0.26 | 0.38 | 0.74 | 0.81 | 0.7 |
| | RF | 0.22 | 0.25 | 0.72 | 0.85 | 0.7 |
| ImgNet-pretr | MLP | 0.16 | 0.10 | 0.99 | 0.82 | 0.01 |
| | RF | 0.35 | 0.20 | 0.89 | 0.78 | 0.01 |
| Raw data | MLP | 0.22 | 0.16 | 0.99 | 0.82 | 0.001 |
| | RF | 0.84 | 0.41 | 0.96 | 0.80 | 0.001 |

(i) the GT labels $z$ get perfect scores of 1 for all metrics; (ii) the uniform mix representation exposes the key difference between *mixing-based* ($D, C$) and functional-capacity-based ($E$) measures of the *simplicity of the $c$-$z$ relationship*, since it attains very low mixing, but near-perfect explicitness scores; and (iii) larger representations (ImgNet-pretr, raw data) tend to be more explicit than the smaller ones (VAE, $\beta$-VAE), with $S$ and $E$ together capturing this size-explicitness trade-off.

## 7 DISCUSSION

**Why connect disentanglement and identifiability?** Connecting prediction-based evaluation in the disentanglement literature to the more theoretical notion of identifiability has several benefits. Firstly, it provides a concrete link between two often-separate communities. Secondly, it endows the often empirically-driven or practice-focused disentanglement metrics with a solid and well-studied theoretical foundation. Thirdly, compared to the commonly-used MCC or Amari distance, it provides the ICA or identifiability community with more fine-grained empirical measures.

**Measuring probe capacity.** Our measure of explicitness $E$ depends strongly on the choice of capacity measure for a probe or function class. For some probes, e.g. random forests or random Fourier features (Rahimi and Recht, 2007; Belkin et al., 2019), there exist natural measures of capacity. However, for other probes like MLPs, coming up with a good capacity measure is

itself an important and active area of research (Jiang* et al., 2020; Dziugaite et al., 2021). While we used the number of parameters to measure the *available* (or upper-bound) probe capacity, future work may improve the soundness of $E$ by leveraging recent (and future) advances in MLP capacity measures, e.g. those which measure the *used* or *effective* capacity of a trained MLP (Hanin and Rolnick, 2019; Maddox et al., 2020).

**Measuring feature importance.** Similarly, the choice of feature-importance measure has a strong influence on the $D$ and $C$ scores, with some probes having natural or in-built measures (e.g. random forests) and others not (e.g. MLPs). For the latter, we proposed the use of probe-agnostic feature-importance measures like SAGE or SHAP (Covert et al., 2020; Lundberg and Lee, 2017), and specified the conditions (Corollary 3.5) that importance measures must satisfy if the resulting $D$ and $C$ scores are to be connected to identifiability. As with probe capacity, coming up with good measures of feature importance is its own orthogonal field of study (e.g., model explainability), with future advances likely to improve the DCI-ES framework.

**What is the relation to causality?** Like the DCI framework and other disentanglement evaluations based on *observational* data, the $D$ and $C$ scores assume statistically independent factors $z_j$, corresponding to a rather trivial causal graph. To relax this assumption and compute $D$ and $C$ in the presence of dependencies, one option is to resort to *interventional* data: Suter et al. (2019) allow for unobserved confounding and propose a measure based on robustness to interventions $do(z_j)$ on the ground-truth factors (see their Defns. 2&3), which can also be used to construct a matrix of feature importances $R$ (see their Figs. 8–12). Similar ideas may help extend the DCI-ES framework to the evaluation of *causal representations* (Schölkopf et al., 2021; Schölkopf and von Kügelgen, 2022).

**What about explicitness for other tasks $y$?** While we focused on the explicitness of a representation for predicting or recovering $z$, one may also be interested in its explicitness for other tasks or labels $y$. While it is often implicitly assumed that ease of predicting $z$ correlates with ease-of-use for other common tasks of interest (e.g., object classification, segmentation, etc.), future work could directly evaluate the explicitness of a representation for particular tasks $y$. For example, one could consider the entire loss-capacity curve when benchmarking self-supervised representations on ImageNet, rather than just linear-probe performance (a single slice). In addition, one could explore the trade-off between explicit but task-specific representations and implicit but task-agnostic representations.

**Conclusion.** We have connected DCI scores to identifiability, and presented an extended DCI-ES framework which introduces two new complementary measures and elucidates how probe-agnostic measures of feature importance can be employed to compute the $D$ and $C$ scores for arbitrary black-box probes. In particular, we advocated for additionally measuring the explicitness E of a representation by the functional capacity required to use it, and proposed to quantify this explicitness using loss-capacity curves. Together with the relative size S of a representation, we believe that DCI-ES constitutes a more fine-grained and nuanced evaluation of representation quality.

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
