# OpenReview forum: "On the DCI Framework for Evaluating Disentangled Representations: Extensions and Connections to Identifiability"
_auai.org/UAI/2022/Workshop/CRL — CRL@UAI 2022 Poster_

### Official Review · Reviewer_L6eM · 2022-06-24
**The paper presents an interesting extension of DCI framework for evaluating disentanglement.**

**Rating:** 6
**Confidence:** 3

**Review:**

The paper considers to extend the DCI framework by considering the functional capacity. Using blackbox predictors, the authors proposes a new score named DCI-ES for better evaluating the quality of the learned features. Experiments on MPIP-3D dataset show that ES scores help to achieve a more complete evaluation on the representations.
My question is on the definition of loss capacity. It seems that the definition of 4.1 is not computable since it needs to compute the supreme. To what extent does the “sampling” procedure approximate the quantity of interest? This is important for establishing the theoretical soundness of the whole method.

---

### Official Review · Reviewer_wmMb · 2022-06-27

**Rating:** 6
**Confidence:** 4

**Review:**

### Summary
This paper draws connections between the DCI metric developed in the disentanglement community and identifiability results which emerged from the nonlinear ICA community. It also propose complementing DCI with _explicitness_ (E), which measures how much capacity is required to recover the ground-truth latents from the learned ones, and _size_, which measure the relative dimensionality of the ground-truth representation over the learned one.

### Review
I enjoyed reading this manuscript and I believe the connection it draws between DCI and identifiability theory is important and will be of interest to the attendees of this workshop. The additional E and S metrics proposed also seems valuable. I thus recommend acceptance. I give only 6 because I believe many important points should be clarified:

I am not sure I can put in words what is the difference between _informativeness_ (I) and the newly proposed E. It seems that, just like E, informativeness is dependent on the choice of probe. Maybe the connection between the two could be clarified? Also, given how sometimes arbitrary measures of capacity can be, I'm not sure I see the value of reporting the AUC instead of simply the informativeness for a few choice a probes with various architectures/hyperparameters. Also, when computing E, how is the smallest capacity value selected? It seems like it could greatly affect the metric's value.

I found Definition 2.1 a bit vague. I suggest adding some example of choice of R close to the definition to make things more concrete.

In the paragraph just before Section 3: "The D and C metrics are only informative when the
data-generating factors z_j are mutually independent." The same point is repeated in the discussion. I think this should be expanded on. Why is it the case? I believe this actually depends on the specific choice of R, no? For example, if R = |W| like in Corollary 3.4, I think D and C are still meaningful even if the z's are not mutually independent. But, if R is the Pearson correlation matrix (in absolute value) between the ground-truth and learned representation, then I do believe D and C could be misleading indeed.

---

### Meta-Review · Program_Chairs · 2022-07-05

**Recommendation:** Accept (Poster)
**Confidence:** 3

**Metareview:**

Both reviewers pointed out that the paper is very interesting in finding connections between DCI and identifiability in nonlinear ICA, and provided a list of suggestions to improve it even further.

---

### Decision · Program_Chairs · 2022-07-06

Accept (Poster)